# Endosymbiotic Fungal Diversity and Dynamics of the Brown Planthopper across Developmental Stages, Tissues, and Sexes Revealed Using Circular Consensus Sequencing

**DOI:** 10.3390/insects15020087

**Published:** 2024-01-29

**Authors:** Yichen Cheng, Jing Yang, Tianzhu Li, Jiamei Li, Meng Ye, Jing Wang, Rongzhi Chen, Lili Zhu, Bo Du, Guangcun He

**Affiliations:** 1State Key Laboratory of Hybrid Rice, College of Life Sciences, Wuhan University, Wuhan 430072, China; 13872209530@163.com (Y.C.); 2022102040037@whu.edu.cn (J.Y.); 2018202040070@whu.edu.cn (T.L.); 2022202040061@whu.edu.cn (J.L.); 2021202040085@whu.edu.cn (M.Y.); whuwangjingxz@163.com (J.W.); rzchen@whu.edu.cn (R.C.); zhulili58@sina.com (L.Z.); 2Hubei Hongshan Laboratory, Wuhan 430070, China

**Keywords:** brown planthopper, circular consensus sequencing, endosymbiont, rice, yeast-like symbiont

## Abstract

**Simple Summary:**

The brown planthopper (BPH, *Nilaparvata lugens* Stål) is the most destructive pest in rice. Understanding the endosymbiont communities hosted by this insect is a prerequisite for controlling BPH rice infestations. This study reveals the diversity and dynamics of endosymbiotic fungi in the developmental stages and various tissues of BPH by means of circular consensus sequencing (CCS), and we found three dominant yeast-like symbionts (YLSs) in almost all samples and the significant biomarkers between female and male adults. This suggests that these endosymbionts are necessary for BPH survival. This study provides a comprehensive survey of symbiotic fungi in the BPH and offers new strategies for pest management in rice.

**Abstract:**

Endosymbiotic fungi play an important role in the growth and development of insects. Understanding the endosymbiont communities hosted by the brown planthopper (BPH; *Nilaparvata lugens* Stål), the most destructive pest in rice, is a prerequisite for controlling BPH rice infestations. However, the endosymbiont diversity and dynamics of the BPH remain poorly studied. Here, we used circular consensus sequencing (CCS) to obtain 87,131 OTUs (operational taxonomic units), which annotated 730 species of endosymbiotic fungi in the various developmental stages and tissues. We found that three yeast-like symbionts (YLSs), *Polycephalomyces prolificus*, *Ophiocordyceps heteropoda*, and *Hirsutella proturicola*, were dominant in almost all samples, which was especially pronounced in instar nymphs 4–5, female adults, and the fat bodies of female and male adult BPH. Interestingly, honeydew as the only in vitro sample had a unique community structure. Various diversity indices might indicate the different activity of endosymbionts in these stages and tissues. The biomarkers analyzed using LEfSe suggested some special functions of samples at different developmental stages of growth and the active functions of specific tissues in different sexes. Finally, we found that the incidence of occurrence of three species of *Malassezia* and *Fusarium* sp. was higher in males than in females in all comparison groups. In summary, our study provides a comprehensive survey of symbiotic fungi in the BPH, which complements the previous research on YLSs. These results offer new theoretical insights and practical implications for novel pest management strategies to understand the BPH–microbe symbiosis and devise effective pest control strategies.

## 1. Introduction

Endosymbiotic microorganisms are widely present in plants and animals. They play important roles in their hosts, such as in nutrient metabolism, growth, development, and resistance to abiotic invaders [1,2,3]. The diversity and dynamics of these microbial communities depend on multiple factors, including their geographical origin, developmental stage, sex, and tissue. For instance, the larva, nymph, adult male, and adult female of the tick *Ixodes pacificus* (Ixodida: Ixodidae) from different locations have significantly different microbial compositions; it has been demonstrated that ticks acquire a significant proportion of their microbiome through environmental exposure [4]. Some studies have shown that the bacterial diversity and function in mosquitoes are associated with developmental stage, sex, and ecological factors, and they may play critical roles in the nutrition, growth, and reproductive functions of their hosts [5,6]. In addition, studies on the diversity and composition of bacterial communities in the head, gut, and ovary after tetracycline treatment found that antibiotic exposure continued to affect bacterial symbionts in different body parts of the planthopper *Laodelphax striatellus* (Hemiptera: Delphacida) [7].

The brown planthopper (BPH), a monophagous pest, has become the most destructive rice pest in the Asia–Pacific region in recent years. At present, control strategies for the BPH mainly rely on chemical insecticides and the development of BPH-resistant rice varieties [8,9]. Recently, the development of biological agents targeting endosymbiotic microorganisms in insects has emerged as a green and efficient strategy for controlling these pests. There are a variety of bacterial and fungal symbionts in the BPH [10,11,12]. The BPH has established a close relationship with yeast-like symbionts (YLSs), endosymbiotic fungi that exist in a huge insect cell, which can participate in the host nitrogen cycle and sterol biosynthesis [13,14,15,16]. In endosymbiotic bacteria, *Wolbachia* can improve BPH reproduction by supplying biotin and riboflavin [17], while *Arsenophonus* bacteria confer B vitamins and insecticide resistance to the BPH [18,19].

The traditional method for the study of insect endosymbionts is to isolate and culture the bacterial and fungal taxa using selective media [20]. However, PCR–denaturing gradient gel electrophoresis (DGGE) technology offers an alternative method to analyze the structure of endosymbiont communities [21]. In recent years, amplicon sequencing technology with the development of high-throughput sequencing has provided a novel method to analyze the community endosymbiont changes in the BPH under different developmental stages, tissues, and treatments [22,23]. However, due to limitations in terms of the low accuracy and short reads of next-generation sequencing, the amplicon sequences could only be annotated at the family or genus level. In some cases, no match to ITS sequences could be found. Circular consensus sequencing (CCS) is a third-generation technology that can produce long reads with an accuracy of over 99.9% [24]. At present, CCS-based full-length amplicon sequencing technology has been widely used in the study of animal endophytes, root microorganisms, environmental management, and insect endosymbionts [25,26,27,28].

In this study, we used CCS to analyze the full-length ITS region of symbiotic fungi at different developmental stages of the BPH (egg, instar nymphs 1–5, female adults, and male adults) and in various tissues of the female and male adult BPH (salivary glands, midgut, Malpighian tubule, hindgut, reproductive organs, fat body, and honeydew). The three YLSs of *Ophiocordycipitaceae* represent the core species in all samples in this study. By comparing the composition and differences in microbial communities, we comprehensively and systematically elucidated the community diversity of symbiotic fungi in the BPH and provide a new direction for exploring novel pest control strategies.

## 2. Materials and Methods

### 2.1. Insects

The brown planthopper was maintained on the susceptible rice variety TN1 and reared in a greenhouse under specific conditions (26 ± 0.5 °C, 70–80% RH, 16 h light/8 h dark) at the Genetics Institute of Wuhan University [29].

### 2.2. Sample Collection

All BPH adults were washed with 75% ethanol for 2 min and rinsed with sterilized ddH_2_O three times. Tissue samples were collected in 1 × PBS (pH 7.4; 140 mmol/L NaCl, 2.7 mmol/L KCl, 10 mmol/L Na_2_HPO_4_, 1.8 mmol/L KH_2_PO_4_) with sterile tweezers under a dissecting microscope from female and male adult BPHs (salivary gland, midgut, Malpighian tubule, hindgut, reproductive organs, and fat body) [30]. Five newly emerged female BPHs were placed in 2.5 cm × 4 cm bags, folded with Parafilm-M, and attached to the base of the rice stem. After the BPH had been allowed to feed on TN1 for 48 h, the bags were removed, and any honeydew was collected with a micropipette [31]. After BPH oviposition, the leaf sheath was washed with 75% ethanol and sterilized with ddH_2_O three times, the fresh eggs were then harvested with sterile tweezers [32]. The samples from the different developmental stages: instar nymphs 1–5, female adults, and male adults, were directly collected and washed as described above. Except for the female midguts, all other samples contained five independent biological replications. All samples were immediately placed in liquid nitrogen after collection and stored at −80 °C.

### 2.3. BPH Virulence Identification

Newly emerged female and male BPH adults were collected and weighed with an electronic balance. Female and male BPH adults with an initial weight of 1.8 mg–2.7 mg and 1.0–1.5 mg, respectively, were used for the experiments. Each selected BPH was placed in a 2.5 cm × 4 cm pre-weighed bag, folded with Parafilm-M. After 48, the body weights and bags were measured again to determine the increased weight of the BPH body and honeydew.

### 2.4. DNA Extraction and PCR Amplification

Fungal DNA was extracted from the BPH samples using the E.Z.N.A.^®^ Tissue DNA Kit (Omega Bio-tek, Norcross, GA, USA) following the manufacturer’s protocols. Using the primers ITS1F 5′-CTTGGTCATTTAGAGGAAGTAA-3′ and ITS4R 5′-TCCTCCGCTTATTGATATGC-3′, where the barcode is an eight-base sequence unique to each sample, the ITS region of the fungi was amplified by using PCR (94 °C for 2 min, followed by 28 cycles at 94 °C for 30 s, 55 °C for 30 s, 72 °C for 60 s, and 72 °C for 5 min). A total of 20 μL mixes containing 4 μL of 5 FastPfu Buffer, 2 μL of 2.5 mM dNTPs, 0.8 μL of each primer (5 mM), 0.4 μL of FastPfu Polymerase, and 10 ng of the template DNA were used to conduct the PCR reactions in triplicate. Using the AxyPrep DNA Gel Extraction Kit (Axygen Biosciences, Union City, CA, USA), amplicons were extracted from 2% agarose gels and purified in accordance with the manufacturer’s instructions.

### 2.5. Library Construction and Sequencing

The amplified DNA was blunt ligated to create SMRTbell libraries as per the manufacturer’s instructions (Pacific Biosciences, Menlo Park, CA, USA). The CCS used in this study is a novel method for generating highly accurate long reads, also known as HiFi reads, from circular DNA molecules. It is based on the principle of circular consensus sequencing, which means that each DNA molecule is sequenced multiple times by a single polymerase as it loops around the circular template. The multiple subreads are then aligned and corrected to produce a single HiFi read with high accuracy and a long length [24]. On dedicated PacBio Sequel II 8M cells, the purified SMRTbell libraries from the Zymo and HMP mock communities were sequenced using the Sequencing Kit 2.0 chemistry (Pacific Biosciences, Menlo Park, CA, USA). On a single PacBio Sequel II cell, purified SMRTbell libraries from the pooled and barcoded samples were sequenced. All amplicon sequencing was performed by Shanghai Biozeron Biotechnology Co., Ltd. (Shanghai, China). 

### 2.6. Statistical and Bioinformatics Analysis

PacBio raw reads were processed using the SMRT Link Analysis software version 9.0 to obtain demultiplexed CCS reads with the following settings: minimum number of passes = 3 and minimum predicted accuracy = 0.99. Raw reads were processed through SMRT Portal to filter the sequences for length and quality. The sequences were further filtered by removing the barcode, primer sequences, and chimeric sequences if they contained 10 consecutive identical bases.

The high-quality sequences were clustered into the same OTUs using UPARSE (version 7.1, http://drive5.com/uparse/, accessed on 1 January 2023) with a 98.65% similarity cutoff, and chimeric sequences were identified and removed using UCHIME (version 4.2, http://drive5.com/uchime/, accessed on 1 January 2023) [33,34]. Venn diagrams were drawn using the online tool ‘Draw Venn Diagram’ (http://bioinformatics.psb.ugent.be/webtools/Venn, accessed on 1 October 2023) to analyze overlapping and unique OTUs during the treatment processes. Each sequence was annotated by the RDP module (https://qiime2.org/, accessed on 1 January 2023) and compared to the Silva database (http://www.arb-silva.de, accessed on 1 December 2022), with a threshold of 80% which means that only sequences that had at least 80% identity (or similarity) to a reference sequence in the Silva database were considered for the comparison [35].

Alpha diversity analysis based on Mothur (version 1.35.1, https://mothur.org/, accessed on 1 March 2015) was conducted to reveal the diversity indices, including OTU abundance, Chao1, ACE, Shannon, and Simpson [36]. The UCLUST algorithm (version 11, https://drive5.com/usearch/, accessed on 1 January 2023) was used to analyze the OTU representative sequences, and the community composition of each sample was counted at each taxonomic level. The alluvial plots were drawn using the R package ‘ggalluvial’ (version 0.12.5, https://cran.r-project.org/package=ggalluvial, accessed on 22 February 2023) [37]. Principal coordinate analysis (PCoA) using the Bray–Curtis distance algorithm and permutational analysis of variance (PERMANOVA) were performed using the R vegan package (version 2.6-4, https://cran.r-project.org/package=vegan, accessed on 11 October 2022) to test the microbial community structures [38,39]. LEfSe (linear discriminant analysis effect size) analysis was performed using the Kruskal–Wallis sum-rank test to screen the biomarkers with LDA values greater than 3, and the cladograms were drawn using Biozeron Cloud Platform LEfSe module (http://www.cloud.biomicroclass.com/CloudPlatform/SoftPage/LF2, accessed on 1 January 2023). Based on the top 30 species, heatmaps were drawn with ‘ggplot2’ (version 3.4.4, https://ggplot2.tidyverse.org, accessed on 12 October 2023) in R to show the differences. The Mann–Whitney U test was used to screen for species with significant differences between males and females. The T-test and Tukey’s HSD test were conducted using SPSS (version SPSS Statistics 29, https://www.ibm.com/spss, accessed on 1 September 2022), with a significance threshold of *p* < 0.05.

## 3. Results

### 3.1. General Profile of ITS Amplicon Sequencing

In order to investigate the composition and microbial communities in different developmental stages and various tissues of the BPH, we generated 1,286,467 raw reads using the Pacbio Sequel II platform. Full-length fungal ITS amplicons were targeted, with an average length of 693.78 ± 94.97 bp (Table 1). After filtering single sequences and chimeras, 740,854 valid sequences were obtained from all samples, including 299,527 sequences from different developmental stages of the BPH, 220,594 sequences from the various tissues of female adult BPHs, and 220,733 sequences from the various tissues of male adult BPHs. Based on the 98.65% similarity cutoff, a total of 87,131 fungal OTUs were obtained, and the number of OTUs in each sample ranged from 4818 to 8910 (Table 1). Among them, 1071, 126, and 541 fungal core OTUs were obtained from the tissues of female adults, the tissues of male adults, and BPHs at different developmental stages, respectively (Appendix A). The presence of honeydew samples reduced the number of shared OTUs in the female tissues, and the number of shared OTUs in the female tissues returned to 587 after removing the honeydew samples (Appendix A). The Shannon–Wiener curves for the three comparison groups almost reached the saturation plateau, indicating that the sequencing depth could cover the major fungal community in the BPH (Appendix A).

### 3.2. The Composition of Fungal Communities

To obtain the species classification information, each OTU was annotated by the RDP module (https://qiime2.org/, accessed on 1 January 2023) and compared to the Silva database (http://www.arb-silva.de, accessed on 1 December 2022), with a threshold of 80%. The total fungal OTUs in all samples were annotated to 5 phyla, 31 classes, 73 orders, 146 families, 336 genera, and 760 species, and the OTU classification taxonomy of each sample at each taxonomic level is shown in Table 1. For a better comparison of the differences in microbial composition and diversity, all samples were divided into three comparison groups, representing the different developmental stages of the BPH, and the female and male adult BPH tissues.

In the samples of the BPH at different developmental stages, *Ascomycota* was dominant at the phylum level, with a maximum of 99.47% ± 0.31% in female adults and a minimum of 79.13% ± 3.60% in male adults (Appendix A). At the genus level, *Polycephalomyces* was the most abundant genus, followed by *Hirsutella*. It is noteworthy that *Malassezia* was the second most abundant genus in the eggs, whereas *Acremonium* and *Moesziomyces* were the second and third most abundant genera in male adults (Appendix A). The trends in composition were more intuitively reflected at the species level, as shown in Appendix A and Figure 1A. *Polycephalomyces prolificus* was found to gradually increase with age, while *Ophiocordyceps heteropoda* gradually decreased with age. *Metarhizium minus* was specifically enriched in the second and third instar nymphs. *Moesziomyces antarcticus*, *Acremonium furcatum*, and *Acremonium* sp. *EN13* were enriched in the male adults. These differences in endosymbiotic fungi prove that the objective characteristics of different developmental stages can significantly affect the abundance and specificity of symbiotic fungi in the BPH. They also indicate the nutritional requirements and functional differences in the BPH at different stages. 

In the various tissues of female adult BPHs, *Ascomycota* was also the dominant fungus (66.25% ± 29.34%~99.26% ± 0.53%) at the phylum level. This was followed by *Basidiomycota* and *Mucoromycota*, with 8.71% ± 14.98% and 2.47% ± 4.68%, respectively (Appendix A). At the genus level, *Ophiocordyceps* and *Polycephalomyces* possessed the highest relative abundance, both of which belonged to *Ophiocordycipitaceae* (Appendix A). According to Appendix A and Figure 1B, the results at the species level were similar to those at the genus level. *Polycephalomyces prolificus* (66.57% ± 6.36%) was significantly enriched in the fat body. However, honeydew, as an in vitro sample, significantly differed from other samples; *Moesziomyces antarcticus*, *Acremonium* sp. *EN13*, and *Pyxidiophora arvernensis* were the dominant species, with 26.62% ± 24.94%, 13.44% ± 10.54%, and 7.22% ± 16.13%, respectively. Notably, *Ophiocordyceps heteropoda* and *Polycephalomyces prolificus* were the most prevalent species in each sample of females except for honeydew, along with *Purpureocillium lilacinum* and *Hirsutella proturicola*, which dominated almost the entire microbial community of the fat body.

The annotation results of different tissues in male adults were similar to those of the females at the phylum level, and *Ascomycota* was the most abundant (69.61% ± 9.17%~91.48% ± 4.46%). The difference was that *Basidiomycota* was present in a much higher proportion than in the females (Appendix A). At the genus level, *Ophiocordyceps* and *Polycephalomyces* were still the most abundant genera, while *Malassezia* was specifically enriched in the male samples (19.48% ± 9.8%) (Appendix A). As is shown in Appendix A and Figure 1C, the high abundance of Malassezia was particularly evident at the species level; *Malassezia restricta*, *Malassezia sympodialis*, and *Malassezia globosa* were significantly enriched in all samples and were more prevalent in the digestive system and reproductive organs than in the fat body. Moreover, *Aspergillus ruber* and *Cladosporium tenuissimum* were significantly higher in the hindgut compared to the other samples. However, the fat body of male adults was still dominated by several common ascomycetes. The tissue specificity of the fungal community of the female and male BPHs provides some new perspectives for studying their phenotypes and behaviors.

### 3.3. Fungal Community Structure Comparisons

The values of the alpha diversity indices of all samples are shown in Table 1. Instar nymphs 1–5 and the male adults have higher richness values, but there is no significant difference between them. The eggs had the lowest richness value, but the highest species diversity value (Figure 2A,B). In the different tissues, the fat body of female adult BPHs possessed the highest Chao1 and ACE values, and there was no significant difference in the Shannon and Simpson indices (Figure 2C,D). The testes and the fat bodies of male adult BPHs had higher richness values, while the salivary gland had the lowest diversity indices. This might indirectly reflect the low virulence of males. Conversely, the high value of the Malpighian tubule might relate to its detoxification ability (Figure 2E,F). In the PCoA analysis, the samples of eggs and male adults were extremely dispersed, and the remaining samples were more concentrated (Figure 3A). Apart from the partial overlap between the Malpighian tubules and hindguts, the differences between the other tissues of female adults were more pronounced, especially in the fat body and honeydew (Figure 3B). Meanwhile, the fat body, testis, and some hindgut samples of male adults were highly dispersed, and others were more concentrated (Figure 3C). Furthermore, the *p*-value of the PERMANOVA analysis was 0.001, which indicated the specificity regarding the diversity of fungal communities in different developmental stages and tissues.

To better compare the species abundance between different groups, and to avoid individual samples not being clustered according to groups due to changes in the abundance of a few species, a heatmap was drawn to more intuitively distinguish the species differences between groups based on the top 30 species (Appendix A). There were 16, 13, and 8 significant species between the samples in their corresponding comparison groups. Three kinds of YLSs, *Polycephalomyces prolificus*, *Ophiocordyceps heteropoda*, and *Hirsutella proturicola*, were dominant in all samples, with significant abundance differences between groups. Moreover, *Malassezia* in the eggs was significantly different from in the other samples, as were *Acremonium* and *Moesziomyces* in the male adults. These differences prove that the objective characteristics of the different developmental stages and various tissues of female and male adults can significantly affect the abundance and specificity of symbiotic fungi in the BPH.

The main microbial communities of the different taxa were analyzed using LEfSe (LDA effect size), which identified biomarkers with inter-group differences based on LDA (linear discriminant analysis) values (Appendix A). To further explore the evolutionary capacity and taxonomic distribution of each host microbial community, evolutionary branching maps of the fungal community were drawn. The taxa that are not significantly different across groups are shown in yellow, while the taxa that are potential biomarkers are colored according to their group association. The biomarkers of female adults in the developmental stage and the fat body of female and male adults were several kinds of YLS, namely, *Ophiocordycipitaceae*, *Polycephalomyces prolificus*, and *Hirsutella proturicola*; their evolutionary branches were relatively close (Figure 4 and Appendix A). Several specifically enriched species showed a significant difference in the eggs and male adults, which were dispersed in the evolutionary branches (Figure 4A and Appendix A). As an in vitro sample, the biomarkers of honeydew are widely distributed and scattered at all taxonomic levels. The Malpighian tubule of male adults as an important organ for detoxification possessed some dispersal biomarkers in their branches, which corresponded to the high Shannon index. These differences might indicate some special functions of the samples at different developmental stages in growth and the active functions of specific tissues in different sexes. 

### 3.4. Comparison of Species Composition and Diversity between Female and Male Adults in Different Tissues

There are some noticeable differences in morphology and feeding ability between male and female BPHs (Figure 5A,B). The significant differences in appearance can be clearly observed from the back, side, and abdomen of the male and female BPH adults. After 48 h, the average weight gain of female and male adults was 1.41 mg and 0.01 mg and the increase in honeydew was 56.30 mg and 3.84 mg, respectively. This indicates that females had a higher feeding ability than males (Figure 5C,D). Although these differences are mainly genetic, they were also closely related to the composition and diversity of symbiotic fungi. Therefore, we compared the species composition and diversity of various tissues of female and male BPHs, including the salivary glands, midguts, Malpighian tubules, hindguts, reproductive organs, and fat bodies.

The comparative analysis of the Chao1 richness index and the Shannon diversity index in the alpha diversity analysis are shown in Appendix A. Among them, only the Malpighian tubule and reproductive organs showed significant differences. PCoA analysis showed that the comparison groups of the salivary glands, hindguts, and ovaries/testes were completely separated by the confidence ellipse, while the other groups partially overlapped (Appendix A).

We further compared the symbiotic fungi of the same organ in male and female BPHs at the species level (Figure 6). As previously described, the fat bodies were dominated by *Polycephalomyces prolificus*, and there was no significant difference between the other groups. *Ophiocordyceps heteropoda* was the dominant species in the other tissues, and its abundance in the female Malpighian tubule, hindgut, and reproductive organs was significantly higher than that of males. Interestingly, the relative abundances of *Malassezia restricta*, *Malassezia sympodialis*, *Malassezia globose*, and the common plant pathogen *Fusarium* sp. in males were higher than those in the females in all comparison groups.

The Mann–Whitney U test was used to examine for differential species across all groups. As is shown in Figure 7, red and blue represent females and males, respectively. Remarkably, *Malassezia restricta* was significantly higher in males than in females in all groups, suggesting its important function. *Purpureocillium* was significantly higher in almost all female tissues compared with that in male tissues. *Polycephalomyces prolificus* in the midgut and *Ophiocordyceps heteropoda* in the hindguts, reproductive organs, and fat bodies was relatively higher in females. 

## 4. Discussion

Symbiotic relationships between insects and endosymbionts play a crucial role in their co-evolution; they are likely to use a strongly imbalanced diet to move into new ecological niches. When insects have only a single nutrient source, these endosymbionts provide the host with nutrients that they cannot produce themselves [40]. Insects provide habitats for endosymbionts, which contribute to host growth, development, reproduction, nutrition, and adaptation [41,42]. For example, when the newly hatched nymphs of the BPH were treated at 35 °C for 3 d, the number of YLSs was significantly reduced, which was calculated using a hemacytometer, resulting in delayed hatching, weight loss, and high levels of uric acid accumulation in the BPHs [14]. In addition, YLSs can provide steroids and more than 10 essential amino acids that the BPH cannot produce on its own, suggesting that YLSs meet the nutritional requirements of the BPH [42]. This study used the CCS technique to obtain a total of 87,131 OTUs, which were annotated to 5 phyla, 31 classes, 73 orders, 146 families, 336 genera, and 760 species. Three dominant yeast-like symbionts (YLSs) were found in almost all samples and the significant biomarkers between female and male adults. YLSs represent the dominant symbiont in the BPH; however, the origin of YLSs is not clear. The 18S rDNA sequence analysis showed that YLSs isolated from the three rice planthoppers originated from *Euascomycetes* rather than *Saccharomycetes* and belonged to *Cordyceps* (*Euascomycetes*: *Hypocreales*: *Clavicipitaceae*) [43,44]. Recently, some studies have characterized the microbial community of the BPH using high-throughput sequencing technology [11,23]. For example, Ren et al. analyzed the amplicon sequencing on the ITS1 region of the BPH at different developmental stages and identified only 269 genera. Of these, *Hirsutella* was the dominant genus; however, they did not describe the endosymbiont at the species level [23]. Traditionally, next-generation sequencing cannot cover the entire 16S/ITS gene region due to the short read length. This usually results in ambiguous taxonomic annotations [45]. Hence, the CCS technique on the Pacbio sequel II platform was used for the first time to investigate the symbiotic fungi of the BPH at the species level. In this study, we identified the three highest relatively abundant YLSs, namely *Ophiocordyceps*, *Polycephalomyces*, and *Hirsutella*, all belonging to *Ophiocordycipitaceae* in *Hypocreales*. This result was different from that of a previous study that identified that the YLSs belonged to *Cordyceps* in *Clavicipitaceae* [44]. It is worth noting that the taxonomic positions of some species of Cordyceps fungi are still unclear. Based on 5~7 gene loci, Sung et al. reclassified a large number of *Cordyceps*-related species into three families: *Cordycipitaceae*, *Ophiocordycipitaceae*, and *Clavicipitaceae* [46]. *Polycephalomyces* and *Ophiocordyceps* share many of the same morphological characteristics, such as *Hirsutella*-like asexual forms and a consistent sexual spore structure [47,48]. Our study found many types of endophytic fungi that have never been previously reported. In addition, our study identified that these fungi in *Ophiocordycipitaceae* are likely to be potential *Euascomycetes*, which is a wonderful supplement to the study of YLSs in the BPH. 

This study first revealed the composition of fungi in honeydew and comprehensively investigated the endosymbiotic fungi in different tissues of the BPH. *Moesziomyces*, *Fusarium*, and *Acremonium* were the most abundant genera in honeydew. *Moesziomyces* can produce free-living saprophytic asexual types (similar to yeast) and plant pathogenic sexual types (smut) [49]. *Fusarium* and *Acremonium* are both widely found in soil and are filamentous yeasts. They are closely related to common pathogens in plants and insects [50,51,52,53]. All of them can produce abundant secondary metabolites, which may contribute to the previously reported microbial-mediated resistance of rice to the BPH honeydew [54,55]. Notably, the proportion of *Malassezia* from *Basidiomycota* was more prevalent in all tissues of males, and *Fusarium* possessed similar trends (Figure 6). *Malassezia* has many biosynthetic genes, including polyketide synthase, non-ribosomal peptide synthase, and terpene synthase, which suggests that *Malassezia* is linked to some male-specific functions [56]. However, the role of microbial diversity in sex differences is unclear, and this may depend on the different physiological structures and life habits of males and females [57]. In summary, these endosymbiont fungi that were found to be specifically enriched in male insects may be derived from their different phenotypes and behaviors. This provides some possible directions for our subsequent research on sexual dimorphism in the BPH.

At present, agricultural antimicrobials have developed into a new type of sustainable pesticide that interferes with the normal life activities of the host by altering the microbial community structure in pests. To date, there have been some studies on the antimicrobial effects of the BPH. Pang et al. used a specific *Arsenophonus* strain to lower the resistance of the BPH to insecticides. This would be useful in the development of strategies to impose an ecological penalty on insect pests [19]. Antibiotics interfered with the bacterial composition in the BPH and suppressed the expression of host detoxifying enzyme genes cytochrome P450 and glutathione S-transferase, thereby affecting insecticide metabolism genes [58]. Shentu et al. used eight different fungicides to significantly reduce the number of YLSs in the BPH, resulting in high BPH mortality [59]. Moreover, the fungicide propiconazole can significantly reduce the total number of YLSs, resulting in a decline in the reproduction and survival rates of the BPH [60]. These results suggest that losing some BPH symbionts affects the original microbial niche. In this study, the microbial diversity and community structure of endosymbiotic fungi in the BPH were characterized in detail. Through the study of the different tissues, different developmental stages, and different sexes of the BPH, we identified several dominant species and several specific species. This has provided new theoretical guidance and practical insights for novel pest management strategies.

## Figures and Tables

**Figure 1 insects-15-00087-f001:**
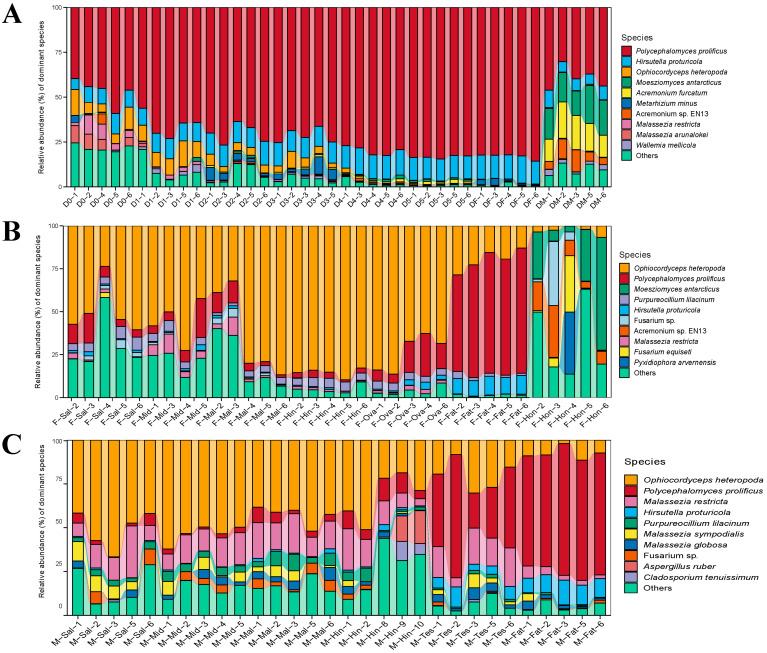
Proportional composition of dominant fungi. (**A**) D0, D1, D2, D3, D4, D5, DF, and DM refer to eggs, the instar nymphs 1–5, female adults, and male adults, respectively. (**B**) F-Sal, F-Mid, F-Mal, F-Hin, F-Ova, F-Fat, and F-Hon refer to the salivary gland, midgut, Malpighian tubule, hindgut, ovary, fat body, and honeydew of adult females, respectively. (**C**) M-Sal, M-Mid, M-Mal, M-Hin, M-Tes, and M-Fat refer to the salivary gland, midgut, Malpighian tubule, hindgut, testis, and fat body of adult males, respectively. ‘Others’ represents the fungi at <1%.

**Figure 2 insects-15-00087-f002:**
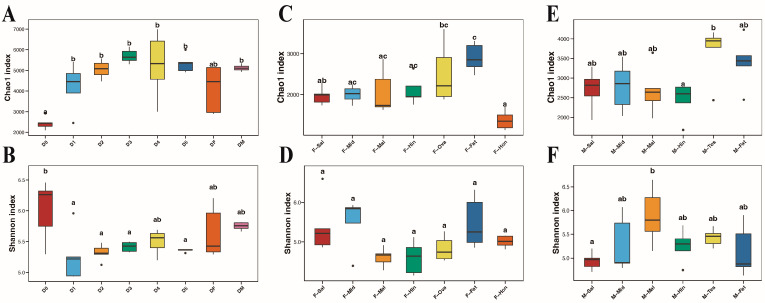
Chao1 (**A**,**C**,**E**) and Shannon (**B**,**D**,**F**) indices of the alpha analysis. D0, D1, D2, D3, D4, D5, DF, and DM refer to the eggs, instar nymphs 1–5, female adults, and male adults, respectively. F-Sal, F-Mid, F-Mal, F-Hin, F-Ova, F-Fat, and F-Hon refer to the salivary gland, midgut, Malpighian tubule, hindgut, ovary, fat body, and honeydew of adult females, respectively. M-Sal, M-Mid, M-Mal, M-Hin, M-Tesm, and M-Fat refer to the salivary gland, midgut, Malpighian tubule, hindgut, testis, and fat body of adult males, respectively. Tukey’s HSD test was used to test the difference between groups. The different letters above each group of samples represent significant differences between groups.

**Figure 3 insects-15-00087-f003:**
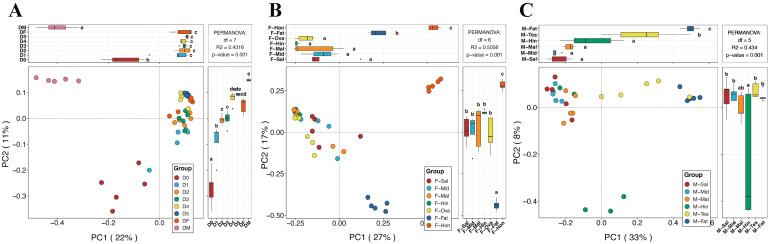
PCoA analysis of Bray–Curtis distances with the statistical test. (**A**) D0, D1, D2, D3, D4, D5, DF, and DM refer to the eggs, instar nymphs 1–5, female adults, and male adults, respectively. (**B**) F-Sal, F-Mid, F-Mal, F-Hin, F-Ova, F-Fat, and F-Hon refer to the salivary gland, midgut, Malpighian tubule, hindgut, ovary, fat body, and honeydew of adult females, respectively. (**C**) M-Sal, M-Mid, M-Mal, M-Hin, M-Tes, and M-Fat refer to the salivary gland, midgut, Malpighian tubule, hindgut, testis, and fat body of adult males, respectively. Tukey’s HSD test was used to test the difference between groups. The different letters above each group of samples represent significant differences between groups.

**Figure 4 insects-15-00087-f004:**
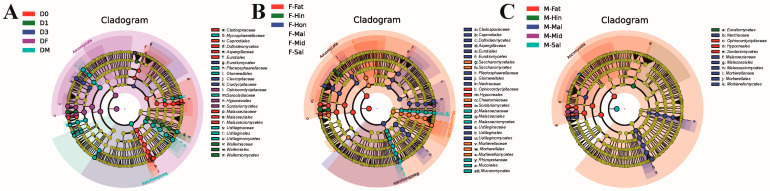
Cladogram of fungi biomarkers in the LEfSe analysis. The small circles in the Figure represent the classifications at different taxonomic levels, and their diameters reflect the relative abundances. The taxa that are not significantly different across groups are shown in yellow, while the taxa that are potential biomarkers are colored according to their group association. (**A**) D0, D1, D2, D3, D4, D5, DF, and DM refer to the eggs, instar nymphs 1–5, female adults, and male adults, respectively. (**B**) F-Sal, F-Mid, F-Mal, F-Hin, F-Ova, F-Fat, and F-Hon refer to the salivary gland, midgut, Malpighian tubule, hindgut, ovary, fat body, and honeydew of adult females, respectively. (**C**) M-Sal, M-Mid, M-Mal, M-Hin, M-Tes, and M-Fat refer to the salivary gland, midgut, Malpighian tubule, hindgut, testis, and fat body of adult males, respectively.

**Figure 5 insects-15-00087-f005:**
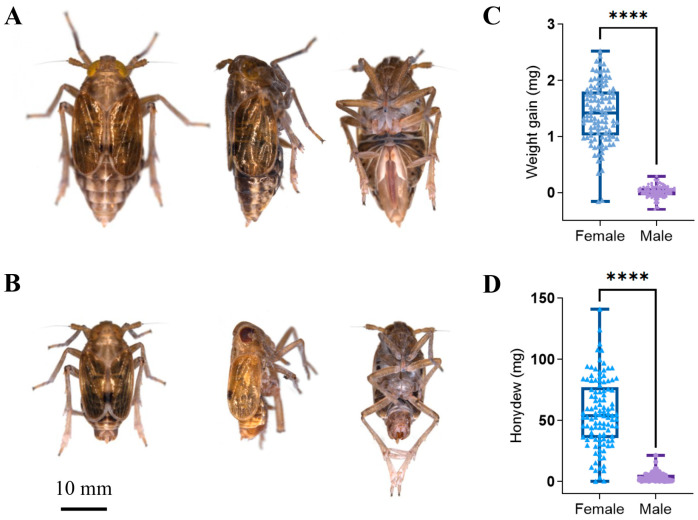
The differences in appearance and virulence of female and male adults. Female (**A**) and male (**B**) BPH adults; bar: 10 mm; the differences in weight gain (**C**) and honeydew (**D**) between newly merged female and male BPH adults, **** *p* < 0.0001 by one-way analysis of variance.

**Figure 6 insects-15-00087-f006:**
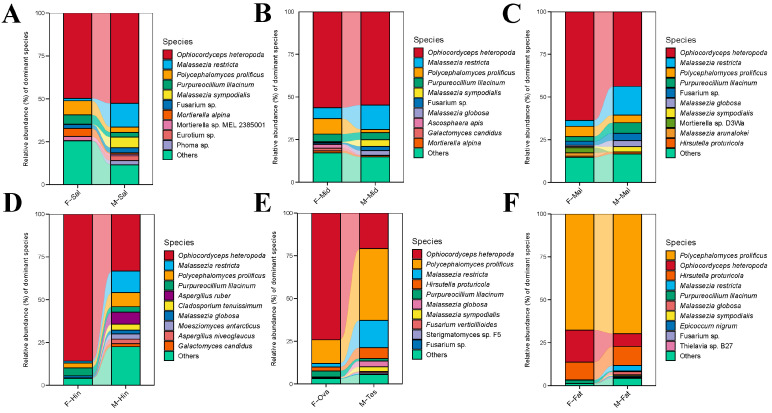
Proportional composition of dominant fungi of different tissues between female and male BPHs at the genus level. F-Sal (**A**), F-Mid (**B**), F-Mal (**C**), F-Hin (**D**), F-Ova (**E**), and F-Fat (**F**) refer to the salivary gland, midgut, Malpighian tubule, hindgut, ovary, and fat body of adult females, respectively. M-Sal (**A**), M-Mid (**B**), M-Mal (**C**), M-Hin (**D**), M-Tes (**E**), and M-Fat (**F**) refer to the salivary gland, midgut, Malpighian tubule, hindgut, testis and fat body of adult males, respectively. ‘Other’ represents the fungi at <1%.

**Figure 7 insects-15-00087-f007:**
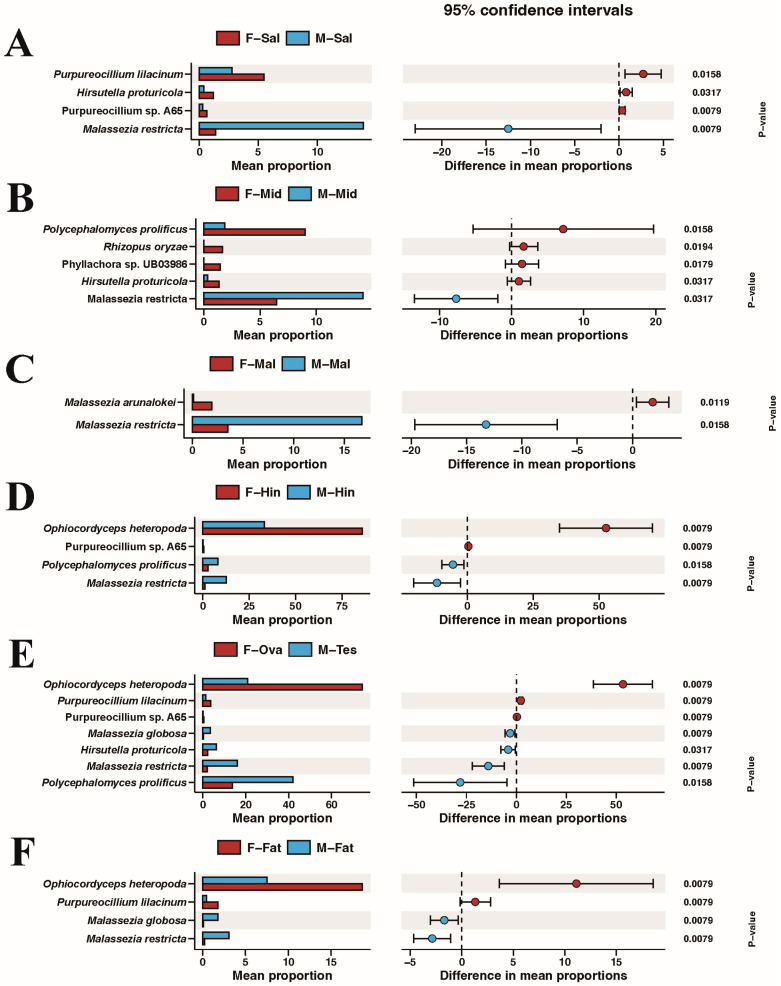
Biomarkers in different groups determined using the Mann–Whitney U test. F-Sal (**A**), F-Mid (**B**), F-Mal (**C**), F-Hin (**D**), F-Ova (**E**), and F-Fat (**F**) refer to the salivary gland, midgut, Malpighian tubule, hindgut, ovary, and fat body of adult females, respectively. M-Sal (**A**), M-Mid (**B**), M-Mal (**C**), M-Hin (**D**), M-Tes (**E**), and M-Fat (**F**) refer to the salivary gland, midgut, Malpighian tubule, hindgut, testis, and fat body of adult males, respectively.

**Table 1 insects-15-00087-t001:** Fungal community alpha diversity characteristics.

Sample	Average Length (bp) ^1^	OTUs	Chao1 ^1^	ACE ^1^	Shannon ^1^	Simpson ^1^
D0	697.18 ± 64.70	8910	2438.46 ± 301.77	2685.45 ± 392.33	6.02 ± 0.49	0.97 ± 0.01
D1	679.50 ± 7.40	7670	4211.15 ± 1128.26	4473.42 ± 1181.89	5.26 ± 0.42	0.91 ± 0.03
D2	682.52 ± 5.24	7376	5050.25 ± 436.99	5387.22 ± 405.28	5.32 ± 0.13	0.91 ± 0.02
D3	682.24 ± 3.11	7920	5694.85 ± 339.04	5991.18 ± 356.93	5.41 ± 0.08	0.91 ± 0.00
D4	722.13 ± 19.83	8548	5258.31 ± 1571.99	5566.48 ± 1568.56	5.50 ± 0.20	0.91 ± 0.01
D5	702.84 ± 7.32	7150	5328.88 ± 428.94	5653.63 ± 434.55	5.35 ± 0.02	0.90 ± 0.00
DF	694.77 ± 1.96	7560	4117.70 ± 1121.43	4410.26 ± 1053.31	5.64 ± 0.41	0.92 ± 0.02
DM	666.93 ± 11.08	7660	5119.20 ± 155.58	5641.36 ± 217.72	5.75 ± 0.06	0.96 ± 0.00
F-Sal	612.42 ± 12.27	7692	1967.43 ± 209.16	2116.07 ± 193.72	5.39 ± 0.70	0.92 ± 0.04
F-Mid	677.73 ± 59.68	7078	2003.80 ± 221.89	2162.75 ± 196.80	5.50 ± 0.74	0.92 ± 0.06
F-Mal	615.61 ± 14.93	4934	2058.24 ± 539.83	2208.23 ± 348.79	4.62 ± 0.23	0.88 ± 0.07
F-Hin	658.89 ± 66.65	6402	2098.50 ± 342.34	2247.86 ± 261.80	4.60 ± 0.41	0.83 ± 0.03
F-Ova	732.75 ± 82.09	5887	2511.27 ± 731.67	2660.79 ± 643.26	4.83 ± 0.31	0.89 ± 0.04
F-Fat	676.28 ± 7.26	7951	2903.54 ± 348.07	3091.26 ± 334.51	5.48 ± 0.64	0.93 ± 0.03
F-Hon	628.93 ± 56.37	4818	1378.55 ± 241.18	1573.77 ± 310.57	5.01 ± 0.15	0.96 ± 0.01
M-Sal	620.45 ± 17.57	5581	2709.21 ± 508.60	2987.41 ± 522.73	4.94 ± 0.18	0.92 ± 0.02
M-Mid	617.28 ± 4.61	7056	2787.83 ± 611.12	3046.64 ± 703.01	5.28 ± 0.58	0.93 ± 0.02
M-Mal	632.90 ± 17.83	8825	2685.96 ± 610.87	2946.71 ± 711.27	5.89 ± 0.59	0.96 ± 0.01
M-Hin	689.25 ± 74.71	8349	2437.76 ± 455.96	2857.79 ± 596.39	5.26 ± 0.35	0.96 ± 0.03
M-Tes	704.38 ± 25.59	8010	3668.89 ± 703.50	4079.21 ± 826.86	5.43 ± 0.18	0.95 ± 0.02
M-Fat	684.36 ± 7.27	7842	3399.84 ± 639.60	3697.81 ± 777.00	5.15 ± 0.54	0.90 ± 0.03
All	693.78 ± 94.97	87,131	3337.92 ± 1455.67	3608.31 ± 1524.12	5.31 ± 0.73	0.92 ± 0.10

^1^ Mean ± standard error. D0, D1, D2, D3, D4, D5, DF, and DM refer to eggs, the instar nymphs 1–5, female adults, and male adults, respectively. F-Sal, F-Mid, F-Mal, F-Hin, F-Ova, F-Fat, and F-Hon refer to the salivary gland, midgut, Malpighian tubule, hindgut, ovary, fat body, and honeydew of adult females, respectively. M-Sal, M-Mid, M-Mal, M-Hin, M-Tes, and M-Fat refer to the salivary gland, midgut, Malpighian tubule, hindgut, testis, and fat body of adult males, respectively.

## Data Availability

All the sequencing data were submitted to the National Center for Biotechnology Information Sequence Read Archive database (NCBI-SRA) under BioProject PRJNA1025130 (SRA accession numbers: SRP464995).

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
