# Peer review of "Endosymbiotic Fungal Diversity and Dynamics of the Brown Planthopper across Developmental Stages, Tissues, and Sexes Revealed Using Circular Consensus Sequencing"

_insects, 2024, doi:10.3390/insects15020087_

Round 1
Reviewer 1 Report
Comments and Suggestions for Authors
The work characterizes in detail the microbial diversity and community structure of endosymbiotic fungal communities in the development stages and various tissues of brown planthopper. The work is interesting and significantly expands scientific knowledge in this area.
However, I have some comments.
Figure 4: The names of taxonomic levels are difficult to read even when magnified on a computer screen, I would suggest increasing the resolution of these graphics.
Comments on the Quality of English LanguageThere are some minor grammatical errors. I would suggest submitting the entire text for proofreading by a qualified person.
Line 268: “the samples of eggs and male adults” instead of “the samples of egg and male adult”
Line 311: “eggs and male adults” instead of “egg and male adult"
Line 346: “different” instead of “defferent”
Line 383: “were treated” instead of “was treated”
Line 394: “genera” instead of “genura”
Reviewer 2 Report
Comments and Suggestions for Authors
The authors conducted a meticulous study of symbionts in BHM, assessing their variations in different tissues, between the two sexes, and across various developmental stages. They employed a novel technique, yielding longer sequences that facilitated species identification. Additionally, they conducted phylogenetic analyses to categorize symbionts into distinct groups.
Despite the significant potential of the work, its English is subpar, making it challenging to comprehend the authors' intended meaning. The manuscript lacks clarity on critical aspects, such as the threshold for species identification based on sequence identity. The authors' evaluation of species quantity is unclear, contributing to uncertainties throughout the manuscript. Both the abstract and summary require a complete rewrite, along with the discussion section.
Several English errors, non-understandable phrases, and incorrectly formatted species names in italics need correction. The term "infection" was used instead of "infestation," which is more appropriate for insects. Notably, results presented in the final part of the introduction should be moved to the results paragraph. The absence of scientific names in the introduction is peculiar.
Greater details on the rearing methodology, including humidity levels, should be provided (lines 99-101). Line 107-109 needs clarification, and line 117-121 requires reorganization due to unclear language. Line 128's sentence needs improvement, as do lines 152-154. The paragraph on statistical analyses contains unfamiliar terms, prompting the recommendation for a review by a subject expert.
Lines 217-219 need clarification regarding whether the sequencing method provides quantitative data on species abundance. Ambiguous sentences (lines 214-223) must be revised to enhance clarity. Figures should be incorporated more synthetically into the text, avoiding redundant information.
Figure 1 requires consistency in color-coding for better comparison. It is advised that species names be written in full the first time, followed by their contracted forms. Lines 246-247 need rephrasing for clarity, and unclear phrases (lines 313-317) should be rewritten.
Figure 6 should maintain consistent color-coding for ease of comparison. In the results section, information often intended for discussion should be appropriately relocated.
Small text size in various figures makes reading difficult, and inconsistent color-coding complicates comparisons. Lines 379-381 require rephrasing for clarity, and lines 383-384 need clarification on the significant reduction of "Yls."
Authors should explain acronyms like "Yls" upon their first use. Additional tables need careful review, ensuring correct use of italics for species names. Ambiguities in table S5 regarding percentage sums not totaling 100% need resolution. Legends or clarifications must address the missing percentage values.
Overall, the manuscript requires significant improvements in language clarity, formatting, and data presentation to enhance its quality and readability.

Comments on the Quality of English LanguageThe quality of English is insufficient, making many sections difficult to understand due to various grammatical errors. Furthermore, I believe it is necessary for an entomologist to review the manuscript because there are several terms that would be more suitable in different scientific contexts.
Certain sections, including the Abstract, summary, and discussions, need complete rewriting. Attached, you will find the PDF file with a selection of sentences and terms that require your review.
Please note that this selection is partial and incomplete
Round 2
Reviewer 2 Report
Comments and Suggestions for Authors
The new version of the ms has improved and I thank the authors for the work done and for the explanations and responses to my requests. In particular, English is really better even if some corrections must be revised.
However, there are still several issues in the ms that must be solved. See below and in the PDF file highlighted in pink.
There are still several errors, grammatical that at least in part I have marked in the pink in the PDF.
The authors must explain the acronyms the first time they quote them, see for example BPH in line 77.
Line 83-85 The authors have just talked about fungi and then talk about Wolbachia and Arsenophonus that are Bacteria, I suggest adding that they are Bacteria in the text.
When authors mention for the first time an insect they must indicated in brackets (Order: Family) then it is necessary to add the author of the species, Moreover, the first time the species must be in full (Laodelphax striatellus). Then after the first-time authors have to use the contracted form (L. striatellus).
Table 1 The authors must explain if “± 64.70” referes to standard error or standard deviation, also in both cases the number of figures after the comma must be an extra figure of the mean value.
This is an example: 5.3 ± 4.70, or 5.34 ± 4,701 and this rule must be used everywhere in the text.
“sp.” It must not be in italics, check in the entire ms.
Figure 4 and 6 have problems, it seems that there is an overlap of two figures. Check please.
There are still in the results of the phrase that must be moved to the discussions. For example, line 388-389 and 392-393, 402, 403; 420-421 But in this case the sentence is also unclear. Please rearrange it.
Line 396 The authors wrote "Ophiocordyceps heteropoda was the dominant genus" but this is the name of a species, check please.
Line 440 please check the number of Orders it seems that 73 197 are too much.
Please put the meaning of EE4 (line 475).
Line 476-477 This sentence is to be reviewed I don't think that exist Fusarium pathogens for human.
Please check.
Please check the numbers of the references in the text in particular the numbers to be mentioned at the 476-477 lines that could be wrong. 54 For example, for example, I think 54 should be cited here. Please check.
The bibliography must be controlled and formatted better.
Please explain the meaning of EE5 Rigo 516. It is likely that it is a problem related to a software for citations. Please Check.

Comments on the Quality of English LanguageThe new version of the ms has improved and I thank the authors for the work done and for the explanations and responses to my requests. In particular, English is really better even if some corrections must be revised.
However, there are still several issues in the ms that must be solved. See below and in the PDF file highlighted in pink.
There are still several errors, grammatical that at least in part I have marked in the pink in the PDF.
The authors must explain the acronyms the first time they quote them, see for example BPH in line 77.
Line 83-85 The authors have just talked about fungi and then talk about Wolbachia and Arsenophonus that are Bacteria, I suggest adding that they are Bacteria in the text.
When authors mention for the first time an insect they must indicated in brackets (Order: Family) then it is necessary to add the author of the species, Moreover, the first time the species must be in full (Laodelphax striatellus). Then after the first-time authors have to use the contracted form (L. striatellus).
Table 1 The authors must explain if “± 64.70” referes to standard error or standard deviation, also in both cases the number of figures after the comma must be an extra figure of the mean value.
This is an example: 5.3 ± 4.70, or 5.34 ± 4,701 and this rule must be used everywhere in the text.
“sp.” It must not be in italics, check in the entire ms.
Figure 4 and 6 have problems, it seems that there is an overlap of two figures. Check please.
There are still in the results of the phrase that must be moved to the discussions. For example, line 388-389 and 392-393, 402, 403; 420-421 But in this case the sentence is also unclear. Please rearrange it.
Line 396 The authors wrote "Ophiocordyceps heteropoda was the dominant genus" but this is the name of a species, check please.
Line 440 please check the number of Orders it seems that 73 197 are too much.
Please put the meaning of EE4 (line 475).
Line 476-477 This sentence is to be reviewed I don't think that exist Fusarium pathogens for human.
Please check.
Please check the numbers of the references in the text in particular the numbers to be mentioned at the 476-477 lines that could be wrong. 54 For example, for example, I think 54 should be cited here. Please check.
The bibliography must be controlled and formatted better.
Please explain the meaning of EE5 Rigo 516. It is likely that it is a problem related to a software for citations. Please Check.
